# Identifying the Translatome of Mouse NEBD-Stage Oocytes via SSP-Profiling; A Novel Polysome Fractionation Method

**DOI:** 10.3390/ijms21041254

**Published:** 2020-02-13

**Authors:** Tomas Masek, Edgar del Llano, Lenka Gahurova, Michal Kubelka, Andrej Susor, Kristina Roucova, Chih-Jen Lin, Alexander W. Bruce, Martin Pospisek

**Affiliations:** 1Laboratory of RNA Biochemistry, Department of Genetics and Microbiology, Faculty of Science, Charles University, Viničná 5, 128 44 Prague 2, Czech Republic; Llano@iapg.cas.cz (E.d.L.); Ronea@seznam.cz (K.R.); 2Laboratory of Biochemistry and Molecular Biology of Germ Cells, Institute of Animal Physiology and Genetics, CAS, Rumburská 89, 277 21 Liběchov, Czech Republic; lveselovska@prf.jcu.cz (L.G.); Kubelka@iapg.cas.cz (M.K.); 3Laboratory of Early Mammalian Developmental Biology (LEMDB), Department of Molecular Biology and Genetics, Faculty of Science, University of South Bohemia, 370 05 České Budějovice, Czech Republic; awbruce@prf.jcu.cz; 4Queen’s Medical Research Institute, MRC Centre for Reproductive Health, The University of Edinburgh, 47 Little France Crescent, Edinburgh EH16 4TJ, UK; Chih-Jen.Lin@ed.ac.uk

**Keywords:** polysome profiling, polysome fractionation, translatome, mouse oocyte, mouse zygote, mouse early embryo, SW55Ti rotor, RNA-seq

## Abstract

Meiotic maturation of oocyte relies on pre-synthesised maternal mRNA, the translation of which is highly coordinated in space and time. Here, we provide a detailed polysome profiling protocol that demonstrates a combination of the sucrose gradient ultracentrifugation in small SW55Ti tubes with the qRT-PCR-based quantification of 18S and 28S rRNAs in fractionated polysome profile. This newly optimised method, named Scarce Sample Polysome Profiling (SSP-profiling), is suitable for both scarce and conventional sample sizes and is compatible with downstream RNA-seq to identify polysome associated transcripts. Utilising SSP-profiling we have assayed the translatome of mouse oocytes at the onset of nuclear envelope breakdown (NEBD)—a developmental point, the study of which is important for furthering our understanding of the molecular mechanisms leading to oocyte aneuploidy. Our analyses identified 1847 transcripts with moderate to strong polysome occupancy, including abundantly represented mRNAs encoding mitochondrial and ribosomal proteins, proteasomal components, glycolytic and amino acids synthetic enzymes, proteins involved in cytoskeleton organization plus RNA-binding and translation initiation factors. In addition to transcripts encoding known players of meiotic progression, we also identified several mRNAs encoding proteins of unknown function. Polysome profiles generated using SSP-profiling were more than comparable to those developed using existing conventional approaches, being demonstrably superior in their resolution, reproducibility, versatility, speed of derivation and downstream protocol applicability.

## 1. Introduction

Protein synthesis is not solely dependent on the steady state level of transcript abundance caused by the trade-off between mRNA transcription and degradation but also upon the dedicated regulation of specific mRNA translation. Indeed, it has already been demonstrated that this level of gene expression is exhaustively regulated and is centrally important in many physiological situations, including stress and during development [1,2,3,4]. 

For almost 50 years, polysome profiling has provided a gold-standard method to precisely assay functional genome readout at a given time point [5]. Indeed, it has for many years been widely applied to assess translational fitness under various physiological conditions relating to cellular stress [6,7,8,9], as well as being applied to study ribosome biogenesis and the functional roles of proteins involved in translation, regulation of mRNA stability and miRNA-mediated silencing [10,11,12,13]. Although, originally established in yeast, polysome profiling has been successfully introduced in mammalian cell, plant, bacteria and translation competent cell-free model systems [14,15,16,17]. In contrast to ribosome profiling, which determines the position of ribosomes within mRNA coding sequence, polysome profiling enables the analysis of entire mRNA transcripts. Consequently, this permits the study of individual transcript isoforms with particular resonance for the identification of conserved *cis*-acting sequences and structural motifs that can participate in transcript-specific translational regulation [18,19,20,21]. Furthermore, polysome profiling is readily combinable with several downstream applications, including microarrays and next generation sequencing (NGS) techniques, to identify polysome-bound mRNAs [16,20,21]. Alternatively, Western blot, proteomic and affinity capture methods can be applied to research proteins specifically associated with ribosomes/polysomes and to study the function of translation initiation factors [22,23,24]. Importantly, the identification of mRNAs engaged in active translation has the inherent potential to aid our understanding of the long-standing observations relating to intra-cellular discrepancies in transcriptome and proteome composition [25,26]. 

The wide-spread utilisation of polysome profiling has nevertheless been historically hindered by its intrinsic complexity, time-consuming nature, limited capacity for high throughput adaptation and its requirement for relatively large initial sample sizes [27]. The most commonly applied protocols utilise sucrose gradients, of varying concentrations, with volumes of ~12 mL, most frequently in the Beckman SW41Ti rotor [7,9,13,28], and the technique has been successfully scaled up using SW28Ti rotor tubes (with 38 mL gradients) to accommodate up to 45 OD_260 nm_ units of cell extract [10,29]. However, efforts to reduce sample size and sucrose gradient volume, more applicable to biological specimens of initial scarcity, have provided limited success [30,31,32]. 

An appropriate progression through meiotic maturation in female germ cells (oocytes) is essential for successful fertilisation and the birth of healthy offspring. Mammalian oocytes undergo two successive cell divisions without an intermediate replicative phase during their maturation. At the onset of meiosis I, the nuclear lamina is phosphorylated and disassembled, leading to nuclear envelope breakdown (NEBD). Concomitantly, the chromosomes condense, and progressive reorganisation of microtubules into a bipolar spindle occurs [33], culminating in the first asymmetric cell division (generating the first polar body), the separation of homologous chromosome bivalents at the end of meiosis I and the generation of a fertilisable egg. However, developing oocytes only actively transcribe their genome during their growth phase, within the ovarian follicles, and store such mRNAs in ribonucleoprotein particles for subsequent translation after the resumption of meiosis and into early embryonic development [4]. Thus, following prolonged arrest in the prophase of meiosis I, there is a unique absence of de novo transcription that means oocytes uniquely and solely rely on the spatial and temporal regulation of mRNA stability and translation for their maturation, thus, making oogenesis a very attractive model system to study RNA related biology. The meiotic maturation of oocytes, within this context, requires a tightly coordinated array of cellular and metabolic processes that includes the inhibition of cell death, remodelling of chromatin and preparation for the metabolic demands connected with the first mitotic division and subsequent fertilisation [34,35]. Accordingly, such processes are largely mediated by the pre-synthesised maternal mRNA stored in translationally-inactive ribonucleoprotein particles [36]. The ribosomal recruitment of specific and required mRNAs during oocyte maturation is controlled by a complex regulatory network involving a central role for the cytoplasmic polyadenylation of targeted transcripts at their 3′-end [37,38,39,40], leading to the creation of translational hotspots (under the influence of the mTOR-eIF4E signalling axis) essential for meiotic maturation [41].

During the past two decades, there has been a shift from the study of individual post-transcriptional mRNAs populations towards high throughput and multiplexed analyses of global mRNA abundance [42]. There are numerous comprehensive studies, from various species including human, reporting temporal patterns of transcriptome composition at various stages of the oocyte development, in zygotes and during early embryogenesis [43,44,45,46]. These experiments have provided valuable insight but have not been able to distinguish stored and translationally active mRNA populations. Therefore, various microarray-based approaches (themselves limited to, and thus biased by, the composite probe features on the array) have been developed to assay polysome-bound mRNAs from limiting numbers of oocytes [30,31,47]. These have latterly been supplemented by RNA-seq based approaches restricted to organisms, where only large number of eggs are accessible, such as the sea urchin and the zebrafish [48,49]. Thus, there exists a requirement for the development and refinement of novel techniques that can both comprehensively assay polysome associated mRNAs and be applied to limiting/scarce amounts of starting material, such as mammalian oocytes [27]. 

Here we provide a detailed protocol, that we have specifically optimised for polysome profiling of a low number of mammalian oocytes (but is adaptable to other experimental paradigms necessitating the use of scarce starting material) and which is compatible with RNA-seq analysis of polysome-bound mRNAs (we have named it ”Scarce Sample Polysome Profiling”, SSP-profiling). Our approach is founded on the utilisation of reduced-volume ultra-centrifugation tubes for sucrose gradient preparation, using the Beckman SW55Ti rotor, and the omission of heterologous cell lysate during the velocity sedimentation procedure. Additionally, we demonstrate the reconstruction of oocyte polysomal profiles by qRT-PCR, as a reliable approach for the visualization of the actual state of translation. As a proof of concept to demonstrate the suitability of SSP-profiling for downstream RNA-seq analysis, we have determined the translatome of mouse oocytes at NEBD. 

## 2. Results

### 2.1. Optimisation of Polysome Profiling in SW55Ti Tubes

Our aim was to conduct a RNA-seq analysis of polysome-bound mRNAs from NEBD-stage oocytes. Therefore, given the comparative scarcity of our oocyte sample and to counter the potential loss of valuable polysomal RNA, we sought to minimise the polysome profiling protocol volumes involved. Conventional polysome profiling protocols typically employ sucrose gradients established in tubes with a size of 14 × 89 mm and a volume of 12.1 mL, compatible with Beckman SW41Ti ultra-centrifugation rotor. However, we optimised ultra-centrifugation conditions to produce high-quality polysome profiles in tubes with a size of 13 × 51 mm and the volume of 5.5 mL (compatible with Beckman SW55Ti rotor), that resulted in a 2.2-fold decrease in gradient volume. During our initial optimisation experiments, carried out using 8 OD_260 nm_ units of HEK-293 cell line lysate as a common type of conventional sample, and employing centrifugal conditions of 47,500 RPM (274,180× *g*) for 75 min, we discovered that high-molecular weight polysomes were lost from the profile (Appendix A). Following further optimisations, that are also included in Appendix A, we established the most suitable centrifugation conditions producing complete polysome profiles, as being 45,000 RPM (246,078× *g*) for 65 min. 

We next sought to test whether the profiles obtained using the reduced volume protocol in the SW55Ti centrifugation tubes were comparable, in terms of peak resolution and amplitude, to those obtained using the conventional SW41Ti method. Moreover, we also wanted to determine whether the SW55Ti-derived profiles exhibited an equal dependency on sedimentation velocity coefficient (S). Accordingly, we employed cell lysates of the yeast strain W303 *ceg1^ts^*, that carries a thermosensitive allele of the guanylyltransferase gene [50]. When grown at the restrictive temperature (37 °C), the W303 *ceg1^ts^* strain is unable add 5′-methyl-guanosine caps to newly synthesised mRNAs leading to an accumulation of uncapped transcripts that inhibits translation initiation and triggers mRNA degradation. This is reflected in a diminution in polysomal peaks in any derived polysome profile. We therefore performed ultra-centrifugation for each lysate variant - HEK-293 cells and the W303 *ceg1^ts^* yeast strain cultivated at either permissive (24 °C) or restrictive (37 °C) temperatures, using both the newly developed SW55Ti-derived (8 OD_260 nm_ units of lysates) and the original SW41Ti-dependent (20 OD_260 nm_ units of lysates) protocols. 

Figure 1 shows the position, number and amplitude of the corresponding peaks agreed well between each pairwise comparison of equivalent lysate. Please note that yeast and human large ribosomal subunits (LSU) have different sedimentation velocity coefficients (25S and 28S, respectively) leading to a comparative shift of the human LSU peak position versus that derived from the yeast LSU and a consequent shift in the comparative positions of the monosomal and polysomal peaks in the polysome profile. Importantly, this shift was clearly visible and highly similar between the profiles obtained using either the SW41Ti- or SW55Ti-based protocols, demonstrating the comparability of the sucrose concentration gradients prepared in both ultra-centrifugation tube variants. The diminution of polysomes in the W303 *ceg1^ts^* lysate obtained from yeast cultured at the restrictive temperature was also similar in both derived polysome profiles, providing extra confidence in the reliability of the newly developed SW55Ti protocol.

We next isolated RNA from the collected fractions (of equal volume) corresponding to the three lysate variants profiled using either protocol. As can be observed, after agarose gel electrophoresis, the presence and relative abundance of either 18S, 28S (25S in yeast lysates) or both rRNA species corresponded well with the position and the amplitude of the corresponding peak in any one particular polysome profile. Moreover, such visualised and fractionated rRNA gel profiles were highly similar for equivalent lysates polysome profiled using the newly developed and low volume (SW55Ti) or the original (SW41Ti) assays (Figure 1 and Appendix A). Therefore, we conclude that polysome profiles obtained in the SW55Ti centrifugation tubes are equivalent to those profiles obtained by the commonly utilised SW41Ti tube-based protocol, thus validating our newly developed low volume polysome profiling assay.

### 2.2. Scarce Sample Polysome Profiling (SSP-Profiling) in SW55Ti-Based Format

To test our Scarce Sample Polysome profiling (SSP-profiling) protocol against scarce biological samples, we disrupted 200 NEBD-stage oocytes in 350 μL of lysis buffer and loaded the obtained lysate onto a 10%–50% sucrose gradient prepared in SW55Ti tubes. Ultra-centrifugation was performed for 65 min at 45,000 RPM at 4 °C. However, Figure 2b details that we did not record any peaks in the derived polysome profile. In parallel, we processed 200 NEBD-stage oocyte lysates that had been treated with 0.5 M EDTA, to induce polysome disassembly into free ribosomal subunits, and although the overall absorbance was higher (due EDTA specific absorbance at 280 nm), we were still unable to detect any polysome profile peaks (Figure 2b). Furthermore, as a centrifugation quality control, we co-processed HEK-293 cell lysates (8 OD_260 nm_ units) as described in Figure 2a. Accordingly, we did obtain a good quality polysome profile that could be disassembled if the lysates were treated with EDTA, as evidenced by the disappearance of polysomal specific peaks and the shift towards individual ribosomal subunits (at positions within the gradient typical for 40S, 60S and 80S peaks). Collectively, these results suggest that it is not possible to record absorbance to generate polysome profiles from a limited number of cells, such as 200 mouse oocytes, even in smaller SW55Ti centrifugation tube sucrose gradients. 

However, the inclusion of the centrifugation control HEK-293 samples is not in itself a fully conclusive quality check. Therefore, to test the veracity of the polysomal fractionation directly, we compared levels of 18S and 28S rRNAs in 10 equal fractions from each HEK-293 or NEDB oocyte obtained profile (±EDTA) by qRT-PCR (Figure 2a,b). Thus, quantification of the 18S and 28S rRNA levels served as an indicator of the positions of the 40S small ribosomal subunit (high 18S rRNA, low 28S rRNA), the 60S large subunit (low 18S rRNA, high 28S rRNA) and the 80S monosome (high both), plus polysomes (characterised by alternating high and low levels of both rRNAs reflecting the increasing number of ribosomes in the polysomal fractions) in the obtained polysome profiles. Importantly, we also generated standard curves (derived after cloning the mouse 18S and the 28S rRNA qRT-PCR amplicons into pCR^TM^4-Topo^TM^ plasmids, note that qPCR primers can detect also human rRNA) to permit absolute copy number calculations in each analysed fraction. A comparison of the obtained HEK-293 polysome profile with the result of the rRNA qRT-PCR fraction screening, indicated a good level of concurrence in the polysome profile readout (Figure 2a), typified by detection of the 40S and the 60S subunits within the third and fourth fractions, respectively. In the case of EDTA-treated HEK-293 cells, we observed maximal abundance of 18S and 28S rRNAs in fractions 2 and 3, respectively, indicating a shift in position of the 40S and 60S peaks by one fraction back in the polysome profile. Crucially, similar qRT-PCR analyses enabled the visualization of the otherwise undetectable absorbance peaks of oocyte polysome profiles (Figure 2b). The maximal rRNA abundances characteristic of the 40S and 60S subunits, were again resident in the fractions 2 and 3, respectively, but this was irrespective of whether NEBD-stage oocyte lysates had been treated with EDTA indicating very low concentration of ribosomal complexes in the oocyte lysate.

qRT-PCR-based absolute quantification allowed the calculation of 18S rRNA copy number in the used lysates. We loaded 190 μL of the HEK-293 lysate with RNA concentration of 1.435 μg/μL. Loaded lysate volume corresponded to 2.8 × 10^7^ cells approximately. The sum of 18S rRNA copies in all 10 fractions was 6.74 × 10^11^ (Figure 2a). In the case of NEBD-stage oocytes, a lysate of 200 cells was loaded in a volume of 300 μL. The 18S rRNA copy number was 3.7 × 10^7^ (Figure 2b). Thus, the copy number of 18S rRNA was 1.82 × 10^4^ higher in HEK-293 sample compared to NEBD-stage oocytes.

We noticed a considerable amount of detected rRNA in the first fraction of the NEBD-stage oocyte polysome profile that contrasted with a significantly lower amount in fraction 1 of the HEK-293 cell line. We therefore decided to test to what extent the NEBD-stage oocyte sample might be contaminated with DNA. We repeated the polysomal fractionation of NEBD-stage oocytes and performed reverse transcription, on RNA isolated from individual fractions reactions, with or without added reverse transcriptase enzyme in parallel, followed by qPCR (Appendix A). Accordingly, we calculated only a minimal copy number of contaminating DNA in the oocyte derived profile, that could not account for amount of 18S/28S rRNAs detected in fraction 1 (detected DNA was more than ten thousand times less abundant than RNA, as calculated for the sum of all 10 fractions). We performed an additional parallel experiment using HEK-293 lysate, with an initial RNA concentration of 1.470 μg/μL that we diluted ten thousand times before performing polysome fractionation. Again, we only detected trace amounts of DNA throughout the whole HEK-293 polysome profile (Appendix A). As we utilised qPCR primer pairs producing only short amplicons, we thus concluded that NEBD-stage oocytes contained a detectable level of rRNA degradation products, which then accumulated at the top of the gradient (fraction 1). Additionally, we discovered that ten thousand times diluted HEK-293 lysate produced polysome profiles with a similar quality to the HEK-293 profiles obtained after application of concentrated lysate on sucrose gradients, as determined by qRT-PCR. Additionally, Appendix A includes the cell numbers and the yields of RNA isolated from the diluted HEK-293 and NEBD-stage oocyte lysates, to assist other researchers with optimisation of the starting number of cells for their own adopted SSP-profiling, when different cell types will be applied.

Additionally, we measured extra and independent biological replicates in order to evaluate the effect of EDTA treatment on the disassembly of polysomes in NEBD-stage oocyte and control HEK-293 cell samples (Figure 2c). Thus, by assaying the abundance of 18S and 28S rRNAs within the obtained fractions we were able to calculate the polysome/non-polysomal ratio (P/NP, i.e., polysomal part; fractions 6–10 versus non-polysomal part; fractions 2–5) of the associated detected rRNAs. The P/NP ratio serves as approximate correlate of the translational activity of the sampled cells. In the rapidly dividing HEK-293 cell line, the calculated P/NP ratios of 18S rRNA and 28S rRNA were equal to 1.29 and 1.87, respectively, indicative of highly active translation, but upon addition of EDTA, the ratio reduced, as anticipated, to 0.2 for 18S rRNA and 0.35 for 28S rRNA. For NEBD-stage oocytes, we calculated the P/NP ratio as 0.85/0.87 (18S rRNA/28S rRNA), indicative of less active translation compared to the HEK-293 cell line, and this reduced in the EDTA-treated control sample to a value of 0.33 and 0.45 (18S rRNA/28S rRNA).

Lastly, we prepared four independent NEBD mouse oocyte derived biological replicates of SW55Ti tube-based polysomal profiles for RNA-seq and analysed them for 18S and 28S rRNA abundance by qRT-PCR (Figure 3b). The 18S and 28S rRNA contents of each individual fraction were plotted as an average and recalculated as a percentage of the given total rRNA abundance in the entire gradient. For comparison, Figure 3a demonstrates a typical qRT-PCR-based polysome profile readout of HEK-293 samples. Figure 3c,d illustrate the results obtained using the same approach applied to 200 mouse zygotes and two hundred two-cell stage embryos, respectively.

The polysome profiles of the NEBD-stage oocytes, zygotes, and two-cell stage embryos were broadly similar, with the 40S and the 60S peak maxima in fractions F2–F3 and F3–F4, respectively, and similar quantities of polysomes as evident from the qRT-PCR-visualised polysome profiles. In HEK-293 cells, the 40S and the 60S peak maxima were shifted to fractions F3–F4 and F4–F5 (as in Figure 2a), probably due to incomparably higher concentration of lysates that were loaded on the gradients. When we calculated P/NP ratios of zygotes and two-cell stage embryos samples and compared them to the P/NP ratio of NEBD-stage oocytes (Figure 2c), we observed a mild continuum in the decrease of detected polysomes from the NEBD, through zygote, to two-cell embryo stages (NEBD P/NP (18S rRNA/28S rRNA) = 0.85/0.87; zygote P/NP (18S rRNA/28S rRNA) = 0.46/0.61; two-cell stage embryo P/NP (18S rRNA/28S rRNA) = 0.39/0.52). However, the observed decrease of P/NP ratios cannot itself serve as a reliable indicator of an overall decrease in total protein synthesis throughout the development. In summary, the important overall conclusion of these experiments, including the three developmental mouse oocyte/embryo stages, was to confirm the applicability of the low volume SW55Ti tube-derived protocol to reliably visualise polysome profiles (using a qRT-PCR based readout) from samples of significant biological scarcity.

### 2.3. RNA-Seq Analysis to Reveal the Translatome of NEBD-Stage Oocytes

The fractionated polysome profiles obtained from NEBD-stage mouse oocytes, represented in Figure 3b, were employed in a RNA-seq analysis of polysome-bound (P) and non-polysome-bound (NP) mRNAs. In this particular RNA-seq experiment, we did not employ normalisation to total RNA since there is no transcription at the NEBD stage [4,51] and also maternal mRNA clearance occurs later in the development [52]. The NP region of the polysome profile, besides ribosomal complexes, contains maternally-stored mRNPs. Thus, comparison of the polysomal and the non-polysomal regions that indeed both contain intact mRNAs protected by proteins (and therefore display higher sedimentation coefficients) provides information about the degree of translation activation/repression of individual mRNA transcripts in oocytes. We excluded fraction 1 of the polysome profile because it contains 1% Triton-X100 that has the potential to influence RNA recovery (thus, being technically distinguished from fraction 2–5). Indeed, RNA degradation products have been reported to accumulate at the top of gradients [53,54]. Consistently, we also detected 18S- and 28S rRNA derived amplicons in the equivalent fraction of NEBD-oocyte polysome profiles (Figure 2b and Figure 3b). Such fractions are also known to be highly enriched with RNPs consisting of short non-coding RNAs with lengths approximating 200 bp ([53]; e.g., tRNAs, as can be seen in the gels in Figure 1, fractions 1–2). For these stated reasons and because the RNA-seq library generation protocol we employed did not recover protein-coding RNAs shorter than 200 bp, the inclusion of fraction 1, provided no technical advantage and it was excluded. However, we acknowledge we cannot completely exclude the possibility that fraction 1 may have contained a small fraction of stored RNPs, that we did not characterise in this study.

Accordingly, we pooled the RNA fractions representing the NP- (fractions 2–5) and P-populations (fractions 6–10) and subjected them to RNA-seq according to protocols described in the “Methods” section. A principal component analysis (PCA) of the RNA-seq data (Figure 4a) revealed that the component 1 (PC1; 28%) clustered the NP and P samples and that the NP samples were less variable in both components (PC2; 21%). Although a correlation of gene expression values showed that there was no profoundly obvious global difference between NP and P datasets (Figure 4b). Out of all 25,574 mRNAs transcribed from autosomes and chromosome X; 11,511 transcripts (with a value of fragments per kilobase per million reads mapped (FPKM) > 0.1) were detected in both NP and P datasets, and further 2152 mRNAs were detected only in the NP with 625 displaying expression unique to the P dataset (Appendix A). 1847 transcripts with FPKM > 50 were found in the P dataset. Furthermore, from the 14,288 mRNAs with FPKM > 0.1 in at least one dataset, 5803 transcripts displayed more than 2-fold enrichment in the NP (NP/P > 2) and 2414 transcripts in the P (P/NP > 2, Figure 4b, Appendix A). When we focused on the 2-fold enrichment of transcripts with FPKM > 1, the numbers changed to 4067 mRNAs enriched in the NP dataset and 1778 mRNAs in the P dataset.

We next employed a ‘Gene Set Enrichment Analysis’ with the aim of gaining a complete overview of the global trends in gene expression in NEBD-stage oocytes. Thus, we utilised mean FPKM values of the P and the NP datasets from all four replicates and surveyed them in respect to their enrichment in either Gene Ontology (‘molecular function’ and ‘biological process’) or KEGG metabolic pathways databases (Figure 4c, full version is provided in Appendix A). We were first interested in the enriched gene categories of the most abundant polysome-bound mRNAs. Furthermore, we determined which gene categories were significantly enriched according to their P/NP ratio, thus giving information about actively translated mRNAs in the NEBD-stage oocytes (Figure 4c). The obtained results from both analyses are thoroughly described in the Discussion section of the article.

We additionally performed DeSeq2 analysis of the NP and the P datasets and obtained a list of 355 transcripts identified as significantly and differentially enriched between NP and P datasets at the level of statistical significance *p* < 0.01 (with log2 of FPKM fold changes ranging from −9.11 to 7.93; Figure 5; Appendix A). To validate the results of the RNA-seq analysis, we selected four genes from this list for qRT-PCR analysis using three additional independent biological replicates of P and NP samples. The selected genes enriched in P fractions according to the RNA-seq data were *Paox* and *Astl* (with P/NP ratios of 7.17 and 5.88, respectively), and the genes enriched in NP fractions were *Upp1* and *Sap30bp* (with P/NP ratios of 0.19 and 0.23, respectively). qRT-PCR results confirmed the direction of enrichment in all four genes, however, the strength of the enrichment was weaker compared to the RNA-seq (Figure 4d, Appendix A). Additionally, to confirm applicability of SSP-profiling to other similarly scarce samples, Appendix A includes a result of a qRT-PCR validation of several zygotic genes that have been selected according to the RNA-seq analysis of 200 zygotes of three independent biological replicates (Appendix A), sequenced by Dr. Ch.-J. Lin, Figure 3c provides qRT-PCR visualisation of the corresponding zygotic polysome profiles).

### 2.4. Sensitivity and Versatility of SPP-Profiling for Conventional Sample Sizes and Compatibility with Downstream Methods

Conventional polysome profiling almost exclusively utilises a SW41Ti rotor-based format. The quality of the polysome profiles prepared using our novel low volume SW55Ti tube mediated methodology (SSP-profiling, Figure 1) convinced us that this approach has the potential to fully replace the traditional SW41Ti format. Therefore, we systematically tested our novel approach in polysome profiling of conventional samples, without a limiting number of cells, to provide experimental guidance for other researchers working with conventional sample sizes. We first focused on determining the sensitivity threshold of SW55Ti-based polysome profiling (Figure 6a). Thus, we utilised a common HEK-293 cell lysate (with an RNA concentration 0.87 μg/μL) and prepared a series of 10%–50% sucrose gradients loaded with lysate volumes ranging from 1 to 8 OD_260 nm_ units that were ultra-centrifuged for 65 min at 45,000 RPM. In parallel, we constructed control polysome profiles in SW41Ti tubes, where 5–20 OD_260 nm_ units of the HEK-293 lysate were applied (1.249 μg/μL). These samples were centrifuged in SW41Ti rotor for 3 h at 35,000 RPM (as per conventionally accepted protocols). The result presented in Figure 6a indicates that the application of just one OD_260 nm_ unit to the novel low-volume-based approach produced a polysome profile with well-defined peaks that enabled the detection of polysomes. Furthermore, based on comparisons with the conventionally derived polysome profiles (obtained using the SW41Ti rotor-based system), we estimated that one OD_260 nm_ unit loaded on in our newly developed approach (utilising SW55Ti rotor compatible tubes) corresponds roughly to 2.5 OD_260 nm_ units applied using the conventional method. These results suggest the novel SW55Ti-based methodology of generating polysome profiles is approximately 2.5 times more sensitive, compared to the established method, when actively translating, fast-growing cells are used.

The application of gradients of varying sucrose concentration is commonly used in cases were enhanced resolution of macromolecular complexes within a specific range of sedimentation velocity coefficients is required (e.g., 40S–80S, polysomes). Therefore, we tested whether SW55Ti rotor compatible tubes, characterised by reduced gradient volume, are suitable for such high-resolution separation. Accordingly, we applied different volumes of HEK-293 cell lysate (RNA concentration–1.435 μg/μL) on the following series of sucrose gradients: 5%–20% (515 μL), 7%–30% (350 μL), 15%–40% (300 μL), 10%–50% (206 μL), 15%–55% (206 μL) and 20%–60% (206 μL). Note that differing volumes of identical lysate sample were loaded onto individual sucrose gradients (of differing concentrations) to ensure an approximately equal amount of material would be retained within the gradient after centrifugation. This is because complexes with higher sedimentation velocity are naturally spun to the bottom of the tube (e.g., polysomes in 7%–30% gradient). Ultra-centrifugation was then performed at 45,000 RPM for 65 min at 4 °C, before 65%-sucrose was used to push gradients into the detector cell. We observed clearly distinguishable polysome peaks and high-quality curves in all the tested gradient variants (Figure 6b). Indeed, the gradient of 5%–20% sucrose separated well the region just before the 40S peak to the first polysome; the 7%–30% gradient the region from the 40S peak to the polysomes containing three ribosomes; the 15%–40% gradient seemed to be suitable for good visualization of the region from the 40S peak to middle-molecular weight polysomes. The most common 10%–50% concentration range provided good resolution from the 40S peak to heavy polysomes. However, the 15%–55% gradient proved to be the most optimal for the separation of very heavy polysomes without any considerable loss of resolution in the region around the 40S peak. The gradient 20%–60% also produced very similar profile, but with reduced resolution in the region between the 40S and the 80S peaks. Overall, we concluded that the novel low volume SW55Ti rotor tube-based protocol is able to readily accommodate various sucrose gradients without compromising the quality of polysome separation.

Many studies aim to confirm the association of proteins of interest with ribosomes and/or polysomes by utilising western blot analysis of fractionated polysome profiles. In Appendix A, we demonstrate the reduced volume SW51Ti tube specific protocol can accommodate enough cell lysate to detect any moderately abundant protein and, in parallel, the KCl concentration used in sucrose gradient ultimately affects the western blot result.

## 3. Discussion

High throughput techniques are widely applied to understand time-dependent global changes in gene expression during oocyte maturation, oocyte-to-embryo transition and early embryogenic development, often with special attention to human infertility [42]. Even though, methodical improvements of NGS methods have led to the establishment of suitable protocols for transcriptomic studies of mammalian oocytes, no such approach is available for deep-sequencing of polysome-bound mRNAs [55,56]. High quality polysome profiling has conventionally required large number of cells to obtain well-defined and interpretable peaks (e.g., for HeLa cells approximately 2–4 × 10^7^). However, such amount of material is not available from specific and scarce cell types, such as mammalian oocytes. Several modified polysome profiling methods have been presented in previous studies, all utilizing DNA microarrays, aiming to circumvent this problem. Potireddy and colleagues [30] described an approach for 160–200 MII-stage mouse oocytes and early embryos, that included pelleting translating polyribosomes through a sucrose cushion in 200 µL-centrifugation tubes, two rounds of cDNA amplification, and subsequent microarray analysis. Another approach, optimised for polysome fractionation of 75 bovine oocytes, entailed the application of incompletely crosslinked heterologous carrier polysomes (prepared from *Drosophila* SL2 cell line), centrifugation in small, SW60Ti centrifugation tubes, two rounds of cDNA amplification and microarray analysis [31]. The final published setup, that was applied to analyse groups of 500–600 mouse oocytes in various developmental stages, is based upon more conventional polysome profiling in 12 mL-, 15%–50%-sucrose gradients, cDNA amplification and microarray hybridisation [47]. Here, we report a protocol comprising centrifugation in small SW55Ti tubes, quantification of 18S and 28S rRNA in each fraction of the derived polysome profile, cDNA amplification and RNA-seq analysis of polysome-bound and non-polysomal bound mRNA populations. We have optimised this approach for a sample size of 200 mouse NEBD stage oocytes (Figure 2b and Figure 3b). Previously, published reports have provided a lot of valuable information concerning the specific recruitment of mRNAs to polyribosomes and have highlighted the irreplaceable role of specific translation to support proper developmental competence of maturing oocytes. However, in SSP-profiling, we have included several methodical improvements, including a reduced sample size (i.e., 200 oocytes) and compatibility with subsequent RNA-seq analyses. Moreover, the omission of heterologous cell derived carriers also prevents contamination of resultant NGS libraries by reads of non-mouse origin and thus sustains sequencing depth. As demonstrated our protocol is functional starting with 200 mouse oocytes, reflecting a reduction of the input material in comparative studies [47] or those utilising larger bovine oocytes [31], but we anticipate that with finer optimisation we may be able to feasibly reduce this number to only 100 oocytes (based on preliminary data).

When employing SWTi55 centrifugation tubes, we focused on adjusting the ultra-centrifugation conditions and checking every step of polysome profiling (Figure 1, Figure 2 and Figure 6 and Appendix A). We suggest that a reconstruction of each individual polysome profile using qRT-PCR of 18S and 28S rRNAs is one of the main methodical improvements for profiling samples of biological scarcity (Figure 2 and Figure 3). In comparison to other published protocols, SSP-profiling allows the determination of the actual rate of translation in each individual profile by rough estimation of the area of the derived polysomal peaks, as well as visualizing the amplitude and positions of peaks corresponding to the 40S and the 60S ribosomal subunits and the 80S monosome. The quantification of absolute rRNA copy number also facilitates the quality control of any derived polysome profiles, in addition to estimating the RNA amount in applied oocyte lysate (Figure 2). Using such methodology, we were able to calculate that the copy number of 18S rRNA was 1.82 × 10^4^ higher in HEK-293 samples as compared to NEBD-stage oocytes (Figure 2). We thus estimate that at least 10,000 oocytes would be required to obtain readable peaks in any oocyte derived polysome profile using the ISCO detector (at its given sensitivity). Clearly, such estimates exclude the possibility of using a conventional, non-qRT-PCR based, method as a huge number of animals would need to be sacrificed in one experiment (given the fact that 20–40 developmentally competent oocytes could be obtained from one female mouse). Taken together, the above stated observations highlight the attractiveness of SSP-profiling protocol, given we can reliably detect polysome profiles from only 200 oocytes.

The combined use of at least four biological replicates of NEBD-stage oocytes, zygotes and two-cell embryos allowed us to validate the reproducibility of the described qRT-PCR-based polysome profiles in other types of scarce/limiting sample (Figure 3). We also investigated NEBD-stage oocyte polysome profiles with and without EDTA treatment (Figure 2c). EDTA disassembles elongating ribosomes and is thus commonly applied for verification of the absence of other complexes containing joined ribosomal subunits in the polysomal regions of the obtained profile. The NEBD-stage oocytes showed a more equal ratio of polysomal and non-polysomal rRNA associated regions of the profile (18S rRNA P/NP = 0.85; 28S rRNA P/NP = 0.87) than HEK-293 cells, demonstrating more moderate levels of translation (Figure 2c). The P/NP ratio of EDTA-treated oocytes decreased to 0.33 and 0.45 for 18S rRNA and 28S rRNA, respectively, indicating that the vast majority of the detected 18S and 28S rRNAs represented polysomes. However, according to the P/NP ratio of EDTA-treated NEBD-stage oocyte, we could not confirm or exclude the possibility that there were no high-molecular-weight stored ribonucleoprotein particles (RNP) that co-sedimented with polysomes and thus contained both rRNA species. It has been shown that maternal RNPs are stored in distinct sub-cellular regions in the oocyte [36]. Indeed, microscopic studies have revealed RNP aggregates at the cortex and in fibrillar cytoplasmic lattices and the latter of which also contains inactive ribosomes [36,57]. However, very little is known about their physical properties, complete protein composition nor their velocity sedimentation coefficients. It is noteworthy that no direct experiments have been reported to exclude the presence of high-weight, yet inactive, ribosome complexes in the fractionated polysomal regions of oocytes so far. EDTA-treatment has been only applied to assess the quality of polysome profiling in oocytes, but only when detecting protein-coding transcripts [31]. Contrarily, we have focused on the simultaneous detection of 18S and 28S rRNAs, an approach that we argue, better reflects the presence of RNP containing inactive ribosomes or their subunits, although we acknowledge a more direct method to assess this would be preferable.

Existing microarray-based studies have focused on the differences in mRNA polysome recruitment during mammalian oocyte maturation [30,31,47]. With the aim of providing new insight into the mechanisms regulating meiosis progression, we applied our own polysomal fractionation method, followed by RNA-seq analysis of polysome-bound and polysome-unbound mRNAs (also containing stored mRNPs) on oocytes at the NEBD-stage. This specific developmental time-point was deliberately chosen as the earliest recognizable developmental point indicative of the end of prolonged meiotic prophase I arrest. Moreover, it is also associated with breakdown of the nuclear lamina, chromosomal condensation and the initiation of meiotic spindle assembly [36]. Maturing oocytes exhibit a high rate of aneuploidy that is associated with increased maternal age and atypically short NEBD-stage duration [58,59]. Indeed, following NEBD, a strong poly(A) signal is known to be present in the vicinity of chromosomes where the meiotic spindle is assembled and certain transcripts exhibit specific nuclear/chromatin localisation [36]. Furthermore, it is speculated that such cellular mRNA localisation might serve as a means to ensure the precise timing of translation of specific proteins needed for meiosis progression [41]. Therefore, revealing the actively translated mRNAs at the NEBD stage has the potential to shed extra light upon such mechanisms related to oocyte aneuploidy and NEBD, that have until now been largely overlooked.

We performed RNA-seq of polysomal and non-polysomal mRNAs in four biological replicates (Figure 3b). Our Principal Component Analysis confirmed polysomal and non-polysome-bound mRNAs clustered away from each other, confirming their heterogonous nature (Figure 4a). In agreement, the conducted Gene Set Enrichment Analysis revealed polysomal enrichment of certain mRNAs, which encoded proteins implicated in an increase of total translation or translation initiation rate, with the potential to contribute to increased polysome formation (GO:0005844, category “polysome”, Figure 4c). We also attempted to confirm that we did not lose some portion of detectable and stored mRNPs by the technically required exclusion of fraction 1 of the fractionated polysome profiles. However, since fraction 1 may only contain mRNPs with a maximum sedimentation coefficient of about 20S, it can be theorised that these mRNPs will preferentially contain only short transcripts of some protein-coding genes. Nevertheless, we assayed for any correlation between transcript length and their calculated P/NP ratio, or any correlation between transcript length and their expression levels, in our derived NP- and P datasets (Appendix A). These analyses did not yield any correlation between these parameters, suggesting no significant bias in the derived data. However, we still cannot be completely sure that we did not lose some mRNPs in the initially excluded fraction 1, but it seems unlikely.

Our study extends the knowledge of gene expression changes during oocyte maturation by focusing on a still unexplored stage of oocyte maturation by polysome profiling. As a result, we found some abundant polysome-associated mRNAs that encode proteins with unspecified function in oocyte maturation, e.g., TAPT1, TUSC3, FBXO38, ZSWIM7 and DPCD (Appendix A). Conversely, when we investigated the overexpressed gene categories, we found our data agrees well with many studies describing the overall metabolic status of the maturating oocyte and in many ways complements the current theories relating to the actual synthesis of regulatory proteins and enzymes [34,60]. For example, from the most abundant polysome-bound mRNAs we identified (Figure 4c, Appendix A), we observed an enrichment of gene categories for pyruvate, glutamine, aspartate, alanine, glycerolipid and l-ascorbate metabolic pathways, and an enrichment of mRNAs for many glycolytic/gluconeogenesic enzymes (e.g., glucose-6-phosphate dehydrogenase). Furthermore, mRNAs encoding subunits of mitochondrial and lysosomal ATPases demonstrated the same trend. These findings agree with the well-documented metabolic status of the maturing oocyte [34]. The main source of energy required for oocyte maturation and resumption of meiosis is known to be pyruvate and glutamine, which are metabolised by the TCA (tricarboxylic acid) cycle and oxidative phosphorylation in mitochondria [61]. Moreover, gluconeogenesis is preferred to glycolysis and the expression of glucose-6-phosphate dehydrogenase is increased along with the activity of the pentose phosphate pathway, which produces pentose bodies for the synthesis of nucleosides [34]. Tubulins and tubulin-associated proteins were also amongst another enriched gene category and it is evident that their synthesis supports correct meiotic spindle formation. Additionally, we observed abundant polysome-bound mRNAs coding for translation initiation factors, rRNA binding proteins (including ribosomal proteins) and number of proteins with a well-established role in meiosis (e.g., DAZL, MOS, Cyclins B1 and B2 and AURKA) in enriched gene categories. It is proposed that active translation in oocytes is partially sustained by amino acids generated by protein degradation, in the context where only six amino acid transport system has been identified [62]. Consistently, we identified proteasome subunit mRNAs to be abundantly enriched in polysomes. Indeed, the ubiquitin-proteasome system is also implicated in the regulation of meiotic progression, fertilisation and zygotic genome activation [63]. Lastly, we observed very high level translation of mRNAs encoding oocyte membrane-associated proteins, which play roles in sperm recognition and the regulation of fertilization; e.g., ZP2, ZP3, Ovastacin and several CCT chaperonins [64].

In the second component of our Gene Set Enrichment Analysis (Figure 4c, Appendix A), we were interested in comparing particular mRNA abundancies in the non-polysomal and polysomal regions of the derived polysome profiles. This comparison permitted the degree of translational activation for individual mRNAs to be determined regardless of their relative abundancy because the non-polysomal pool also contains stored mRNPs. Similar results, focusing on the most abundant polysomal transcripts, were observed. However, we discovered that some of the highly translated mRNAs were in fact still more abundant in the stored mRNA pool. This finding applied particularly to mRNAs for both cytosolic and mitochondrial ribosomal proteins, ATPase subunits and some glycolytic enzymes, indicating that only the minority of these mRNA molecules were translated and that the majority still remained pre-synthesised for later translation. The same trend was also evident for many mRNAs coding for protein regulators of meiotic progression. During oocyte growth and maturation, mitochondria replicate up to numbers between 300,000–400,000 [65]. We found that mRNAs for the mitochondrial cytochrome C oxidase subunits were enriched in polysomes, while mRNAs for the NADH:ubiquinone oxidoreductase subunits were more abundant in the non-polysomal mRNA pool. We observed translational activation of mRNAs for resident endoplasmic reticulum proteins and many proteins with antioxidant activity, such as superoxide dismutase, thioredoxin reductase, glutathione peroxidase, glutathione dehydrogenase and peroxiredoxin. We also detected mRNA encoding lactate dehydrogenase to be abundant in polysomes. Taken together, these facts demonstrate that active translation of such proteins, which are significant for the maintenance of the redox state in the oocyte and buffering mitochondria respiratory activity, occur in the NEBD-stage oocytes [66]. mRNAs for two UDP-galactose and one nucleotide solute career transporters were additionally found to be also enriched in polysomes, demonstrating that their active synthesis supports galactose, pyrimidine and adenine-containing base import to NEBD-stage oocytes. Intriguingly, several mRNAs of the spliceosomal complex were enriched in polysomes, whereas mRNAs for several basal transcription factors were enriched in the non-translated mRNA pool. We found Prpf38A mRNA that encodes the spliceosome protein PRP38 and SRSF1 splicing factor encoding mRNA to be enriched in polysomes (Appendix A). This suggests that transcriptionally silent NEBD-stage oocytes contain pre-synthesised mRNAs for proteins involved in both transcriptional initiation and splicing, but the splicing related mRNAs are translated earlier in development.

During the optimization of this novel SSP-profiling protocol, we also used HEK-293 cell line and W303 yeast strain lysates as controls (Figure 1). The performance of SSP-profiling in regard to these conventional samples, in the SW55Ti rotor, was very favourable (Figure 1, Figure 2a and Figure 6). This SW55Ti rotor is occasionally applied in plant science protocols where polysome profiles are often obtained from crude ribosome pellets after centrifugation of lysates over a sucrose cushion in a Type70Ti rotor [67,68]. The SW55Ti rotor, or the even smaller SW60Ti rotor, have been utilised for polysome profiling in a couple of previous studies but either the relevant authors did not provide evidence of the obtained curves or the profiles were obtained from specialised material and/or by different sucrose gradient mediated methods [69,70,71,72]. Owing to these facts and the general sparsity of SW55Ti-derived polysome profile experiments, we decided to carefully compare polysome profiling in SW55Ti and conventional SW41Ti rotors (SW41Ti represents the most often used rotor size format, but sometimes similar SW40Ti is used). Accordingly, for this comparison, we applied the most often studied biological materials; i.e., mammalian cell lines and yeast cells. The results in Figure 1, Figure 6 and Appendix A suggest that the SW55Ti-based polysome profiles are fully comparable to those conventionally obtained with SW41Ti rotors and that this SSP-profiling protocol is fully compatible with downstream methods. Importantly, the usage of smaller tubes offers economic benefits to the whole polysome profiling methodology, as it not only directly saves on required chemicals (to the extent of ~2.2-fold), but also reduces consumable usage, in terms of the material required for cell growth. We also predict that the three times reduction in centrifugation time, plus the additional shortening of gradient analysis time, will be very attractive for a large number of researchers.

## 4. Materials and Methods

### 4.1. Human Cell Line, Yeast Strains and Mouse Oocyte, Zygotes and 2C-Embryos Isolation and Culture

The human embryonic kidney 293 (HEK-293) cell line was maintained in Dulbecco’s Modified Eagle Medium (DMEM, Gibco, Gaithersburg, MD, USA) supplemented with 10% FBS and 2 mM l-glutamine at 37 °C in humidified atmosphere containing 5% CO_2_. Cells were grown to 60%–70% confluency in 150 mm diameter dishes. Cycloheximide was added to medium at final concentration of 0.1 mg/mL. After 10 min, cells were washed with ice-cold PBS supplemented with cycloheximide (0.1 mg/mL) and then scraped in the presence of 800 µL of Polysome Extraction Buffer (PEB; 10 mM HEPES, pH 7.5; 62.5 mM KCl; 5 mM MgCl_2_; 2 mM Dithiothreitol (DTT); 1% Triton X-100; 100 μg/mL cycloheximide; complete EDTA-free (Roche, Basel, Switzerland, 1 tablet/10 mL); and 40 U/mL Ribolock (Thermo Fisher Scientific, Waltham, MA, USA) and collected in 1.5 mL tubes.

Fifty milliliters of the *Saccharomyces cerevisiae* strain W303 bearing a temperature-sensitive mutation in guanylyltransferase (*ceg1-3^ts^*, [50]) gene were cultivated in YPD medium at 24 °C with shaking. The culture was split into two parts at an OD_660 nm_ of 0.6. The first half was immediately harvested, the second one was mixed with pre-warmed, 52 °C-hot medium and cultivated for 12 h at 37 °C. At the time of harvesting, cycloheximide was added to the medium at final concentration 0.1 mg/mL. After 10 min, the cells were pelleted by centrifugation and washed three times with ice-cold water with cycloheximide (0.1 mg/mL). Dry cell pellets were kept at −80 °C for later use.

Germinal vesicle (GV) oocytes were obtained from 8 weeks old CD1 mice, 46 h after injection with 5 IU pregnant mare serum gonadotropin (PMSG, ProSpec, Rehovot, Israel). Oocytes were isolated at the GV stage (0 h) in M16 medium (Sigma-Aldrich, Darmstadt, Germany) [73], supplemented with 100 µM 3-isobutyl-1-methylxanthine (IBMX, Sigma-Aldrich, Darmstadt, Germany) to prevent nuclear envelope breakdown (NEBD). Selected oocytes were denuded and cultured in M16 medium (Sigma-Aldrich, Darmstadt, Germany) without IBMX at 37 °C in 5% CO_2_. After 80 min of culture, those oocytes that underwent NEBD were selected. Finally, after 3 h post-IBMX, cycloheximide at final concentration 0.1 mg/mL was added to medium and oocytes were cultured for 10 further minutes. NEBD oocytes were then transferred to polyvinyl alcohol (PVA) supplemented with cycloheximide (0.1 mg/mL) and stored at –80 °C in a low-binding retention tubes.

In the case of zygotes and two-cell embryos, 48 h post PMSG injection, mice were further injected with 5 IU of hCG (Pregnyl^®^, MSG, Merck Sharp & Dohme B.V, White House Station, NJ, USA) and placed with males for mating. Zygotes and embryos were collected 23 h and 35 h later, respectively, from oviducts, and moved into M2 medium (Sigma-Aldrich, Darmstadt, Germany), which was supplemented with cycloheximide (0.1 mg/mL). After 10 min of cultivation at 37 °C in atmosphere containing 5% CO_2_, they were harvested and stored at −80 °C. All animal work was conducted according to Act No 246/1992 for the protection of animals against cruelty; issued by Ministry of Agriculture of the Czech Republic, 25.09.2014, number CZ02389.

### 4.2. Preparation of Lysates and Polysomal Profiling

All cell types were lysed in PEB. HEK-293 cells were incubated in lysis buffer for 20 min on ice with occasional vortexing. Cell extracts were centrifuged at 8000× *g* for 5 min at 4 °C. Yeast cells were disrupted using acid-washed glass beads in a mixer mill apparatus MM301 (Retsch, Gmbh, Haan, Germany), set at 30 shakes/second for 3 min, followed by immediate cooling on ice. To disrupt oocytes and their rigid zona pellucida (ZP), zirconia-silica beads were added to the tubes containing 200 NEBD-stage oocytes together with 350 µL of lysis buffer. Tubes were placed in MM301 and shaken in the frequency of 30 shakes/second for 1 min followed by immediate cooling on ice for 3 min. This step was repeated 3 times. Lysates were cleared by centrifugation at 10,000× *g* for 5 min at 4 °C and then cast on sucrose gradients, which were prepared in solution containing 10 mM HEPES, pH 7.5; 100 mM KCl; 5 mM MgCl_2_; 2 mM DTT; 100 μg/mL cycloheximide; Complete EDTA-free (1 tablet/100 mL); and 5 U/mL Ribolock (Thermo Scientific). For polysome dissociation experiments, 100 mM EDTA was added to lysis solution without MgCl_2_ prior to disrupting the cells and the resulting lysate was loaded on sucrose gradients supplemented with the same concentration of EDTA. Sucrose gradients were prepared in Gradient Master^TM^ 108 v5.3 (Biocomp, Fredericton, NB, Canada) with the following programs for SW55Ti tubes with long cap: 5%–20% (*w/v*), one-step; 7%–30% (*w/v*), one-step; 15%–40% (*w/v*), one-step; 10%–50% (*w/v*), twelve-step; 15%–55%, two-step; 20%–60% (*w/v*), two-step. SW41Ti long-cap 10%–50% (*w/v*) sucrose gradients were prepared using the eleven-step program. We adjusted the weights of SW55Ti and SW41Ti tubes with prepared gradients to 6.59 g (SW55Ti) and 15.19 g (SW41Ti), respectively, before loading lysates. This protocol improvement allowed us to load the identical volumes of samples and subsequently helped with fitting the position of the loading peak in the polysome profile curve. Meaning, the SW55Ti tube can accommodate 8 OD_260 nm_ of lysate in maximal volume of 300 μL and SW41Ti tube 20 OD_260 nm_ in volume 520 μL. Where lysate concentration is indicated, it corresponds to RNA concentration in the lysate as calculated. Velocity sedimentation was performed in Optima L-90 Ultracentrifuge (Beckman Coulter, Indianapolis, IN, USA) at 45,000 RPM (246,078× *g*) for 65 min at 4 °C in SW55Ti UltraClear^TM^ tubes (Beckman Coulter) or at 35,000 RPM (210,053× *g*) for 180 min at 4 °C in SW41Ti UltraClear^TM^ tubes (Beckman Coulter). Continuous absorbance monitoring at 280 nm was performed using an ISCO UA-5 detector and ISCO UV absorbance reader (Teledyne, ISCO, Lincoln, NE, USA). 60% sucrose solution was pumped into SW55Ti and SW41Ti tubes with the speed 1.8 mL/minute and 2.2 mL/minute, respectively, using NE-1000 syringe pump (New Era Pump Systems, Inc., Farmingdale, NY, USA). Polysome profile data were collected and processed using Clarity Lite software [29].

### 4.3. RNA Isolation, RNA Electrophoresis and qRT-PCR

For coarse RNA isolation from human and yeast cells (Figure 1), we collected SW55Ti and SW41Ti polysome profiles into fractions of equal volume. One microliter of GeneElute^TM^-LPA (linear polyacrylamide, Sigma Aldrich) was added to each fraction as a co-precipitant. After vortexing, equal volumes of 5.25 M guanidine thiocyanate and isopropanol were added. Samples were precipitated overnight and then RNA was spun at 19,000× *g* for 20 min at 4 °C. After two washes with 75% ethanol, pellets were air-dried and RNA corresponding to SW55Ti- and SW41Ti-derived fractions was dissolved in 20 µL and 30 µL of RNase-free H_2_O, respectively. RNA electrophoresis was performed according to [74] with the exception that loading dye was supplemented with 0.1% SDS. We loaded 15 µL (SW55Ti) and 10 µL (SW41Ti) of dissolved RNA. RNA concentration and yield were determined spectrophotometrically at 260 and 280 nm.

High quality RNA isolation from the HEK-293 cell line, mouse oocytes, zygotes and two-cell stage embryos was performed by collecting 10 fractions, of 0.5 mL each, of SW55Ti-derived polysome profiles. Working on ice, 1 µL of GeneElute^TM^-LPA was added, vortexed and immediately added 1 mL of TriReagent (Sigma Aldrich). After vortexing, 350 µL of chloroform was added, again vortexed and centrifuged at 19,000× *g* for 20 min at 4 °C. Equal volumes of isopropanol were added to RNA-containing phases. RNA isolation was then completed as described above in the case of the coarse RNA isolation.

4 µL of RNA from each fraction were reverse-transcribed using 20 U of M-MuLV Reverse Transcriptase (Thermo Scientific) and 0.3 µg of random hexamer primers in a reaction volume of 20 µL. cDNA synthesis was performed at 25 °C for 10 min and then in 37 °C for 5 min followed by incubation at 42 °C for 1 h and subsequent inactivation at 70 °C for 10 min. qRT-PCR experiments were performed using a LightCycler480^®^ (Roche, Basel, Switzerland) and LightCycler480^®^ SYBR Green I Master mix (Roche). The 10-µL reactions were performed in triplicates. Each reaction contained 2 µL of cDNA and 500 nM gene-specific primers (list of used primers is provided in Appendix A). The amplification protocol was 95 °C for 5 min; 44 cycles of 95 °C for 10 s, 58 °C for 15 s, 72 °C for 15 s; followed by melting curve determination. For absolute qRT-PCR quantification, we created recombinant pCR^TM^4-Topo^TM^ plasmids (Invitrogen, Carlsbad, CA, USA) containing 18S and 28S rRNA PCR amplicons (Appendix A). For qRT-PCR validation of selected transcripts, we took half of dissolved RNA in each collected fraction (10 µL) and created two pools: non-polysomal sample (NP; corresponding to the 40S, the 60S and monosomal peaks, fractions 2–5) and polysomal associated one (P; polysomal region, fractions 6–10). Then, we performed reverse transcription reactions in the final volume 100 µL containing 40 µL of pooled RNA, 2.5 µg of oligod(T_18_) and 1 µg of random hexamer primer and 100 U of M-MuLV Reverse Transcriptase. RT reaction and qPCR conditions were identical as those described above. The relative quantification mode was applied and the mean of 18S and 28S RNA level was used for normalization of NP and P components of the polysome profiles.

### 4.4. RNA-Seq Library Preparation

For RNA-seq, we measured 18S and 28S rRNA content in collected fractions derived from NEBD-stage oocyte polysome profiles and then we pooled fractions as described for the qRT-PCR validation. Pooled RNA was concentrated using RNA Clean & concentrator kit (ZYMO Research, Irvine, CA, USA) and rRNA depleted using Ribozero Gold rRNA Removal kit (Illumina, San Diego, CA, USA). Remaining RNA in each sample was DNAse-treated, reverse transcribed and amplified using the Repli-G WTA single cell amplification kit (Qiagen, Hilden, Germany). cDNA synthesis was primed by a mixture of oligod (T_18_) and random hexamers primers. The amplified cDNA was then processed by the Nextera DNA library prep kit (Illumina) to generate sequencing libraries. Sequencing was performed on HiSeq 2500 (Illumina) with 150 bp paired-end reads.

### 4.5. Bioinformatical and Statistical Analyses

Reads were trimmed using Trim Galore! v0.4.1, checked for sequencing quality using FastQC v0.11.5 and for contamination by mapping to genomes of several model organisms using FastQC Screen v0.11.1, and finally mapped to the mouse GRCm38 genome assembly using Hisat2 v2.0.5 [75]. Gene expression was quantified as fragments per kilobase per million (FPKM) values in Seqmonk v1.40.0 (Manufacturer name, city, state abbreviation if USA or Canada, country). Mapping of RNA-seq reads on mouse genome is accessible in UCSC Genome Browser (https://genome.ucsc.edu/s/lenkagahurova/NAR_Masek_polysome_RNAseq). RNA-seq data have been deposited in Gene Expression Omnibus database under the accession: GSE121358. PCA for transcripts with FPKM >1 was performed using R function (R v3.4.0) within Seqmonk v1.40.0. DeSeq2 (R v3.4.0) was applied to analyse transcripts with differential abundancy in the NP and the P samples. The R code for the translatome analysis is available upon request. qRT-PCR data analyses were performed in the LightCycler 480 software (version 1.5, Roche, Basel, Switzerland) in absolute and relative quantification modes. Gene Set Enrichment analysis was performed by utilising the WEB-based GEne SeT AnaLysis Toolkit (http://webgestalt.org/option.php). Results in Figure 2c and Figure 4d were statistically evaluated by Kruskal–Wallis ANOVA followed by post-hoc Dunn´s multiple comparison test.

## 5. Conclusions

In summary, this study describes SSP-profiling as a novel protocol that is able generate high quality data with all the versatility of currently described and conventional methods. We have demonstrated the suitability of this novel polysome profiling technique by coupling it to RNA-seq screens of polysome associated transcripts from biologically scarce samples, as represented by 200 mouse NEBD-stage oocytes (in the process generating novel and biologically relevant data on a previously unexplored point in mouse oocyte maturation development). Lastly, we have also provided a detailed description of the SSP-profiling experimental technique that we anticipate will aid its adoption in the wider scientific community.

## Figures and Tables

**Figure 1 ijms-21-01254-f001:**
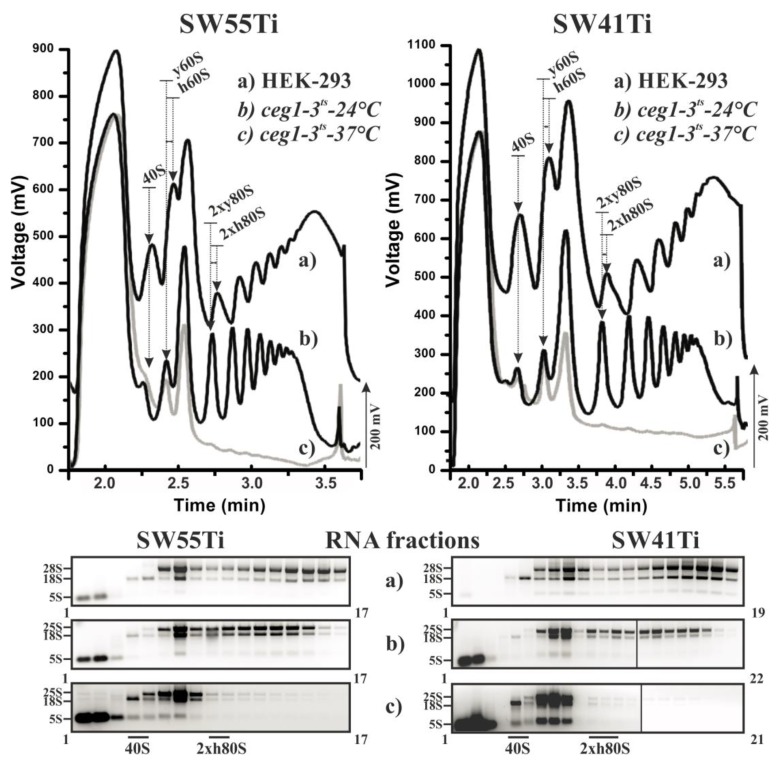
Performance of polysomal profiling in SW55Ti centrifugation tubes. Identical cell lysates from HEK-293 cells (a) and yeast strain W303 *ceg1^ts^* cultivated either at 24 °C (b) or 37 °C (c) were loaded on 10%–50% sucrose gradients prepared either in SW55Ti tubes (left panels) or SW41Ti tubes (right panels). RNA was isolated from equal fractions of the respective profiles and loaded on agarose gel (lower panels). Individual fraction numbers are indicated next to each electrophoretogram. Note in upper polysome profiles the obtained HEK-293 curves are shifted up by 200 mV to prevent overlapping with the yeast derived data curves. Arrows indicate the position of specific peaks within the profiles: i.e., 40S, small ribosomal subunits; 60S, large ribosomal subunits; and 2 × 80S, low molecular weight polysomes consisting of two ribosomes; h denotes human and y signifies yeast ribosomal subunits/polysomes. To the left of each electrophoretogram are denoted the positions of rRNAs in corresponding gels (note, the 5S rRNA label denotes the collective positions of 5.8S rRNA, 5S rRNA and tRNAs in gels).

**Figure 2 ijms-21-01254-f002:**
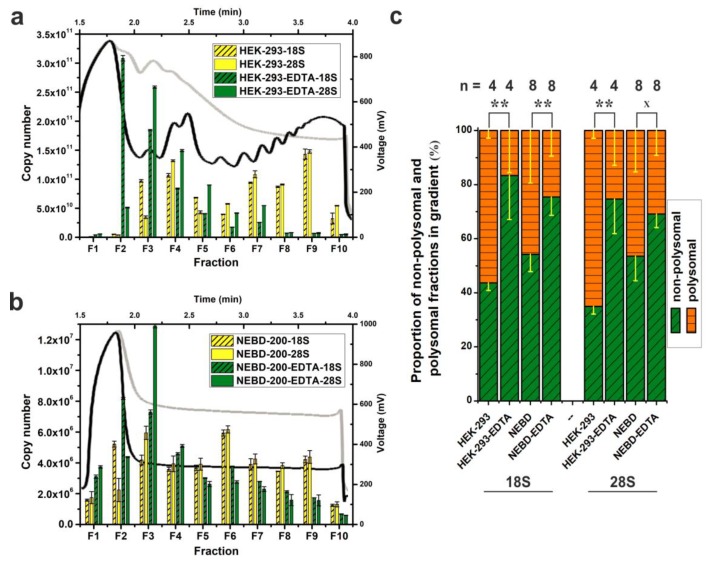
qRT-PCR-based visualization of polysome profiles from scarce samples using SW55Ti tubes. Comparison of polysome profiles of HEK-293 cells (**a**) and 200 NEBD-stage oocytes (**b**). Black curves and yellow columns in each chart indicate untreated lysates; EDTA-treated polysome profiles are traced in light grey and green columns. Absolute copy number of 18S and 28S rRNA was determined in fractions F1–F10 and error bars display ±SD of qPCR technical triplicates. (**c**) Proportion of the non-polysomal (NP, fractions F2–F5, in green) and polysomal (P, fractions F6–F10, in orange) parts of profiles from untreated and EDTA-treated HEK-293 cells and NEBD-stage oocytes; error bars, ±SD; *n*, number of independent biological replicates; ** *p* < 0.001; x signifies a lack of significance; only non-treated and EDTA-treated pairs from Dunn’s multiple comparison are displayed.

**Figure 3 ijms-21-01254-f003:**
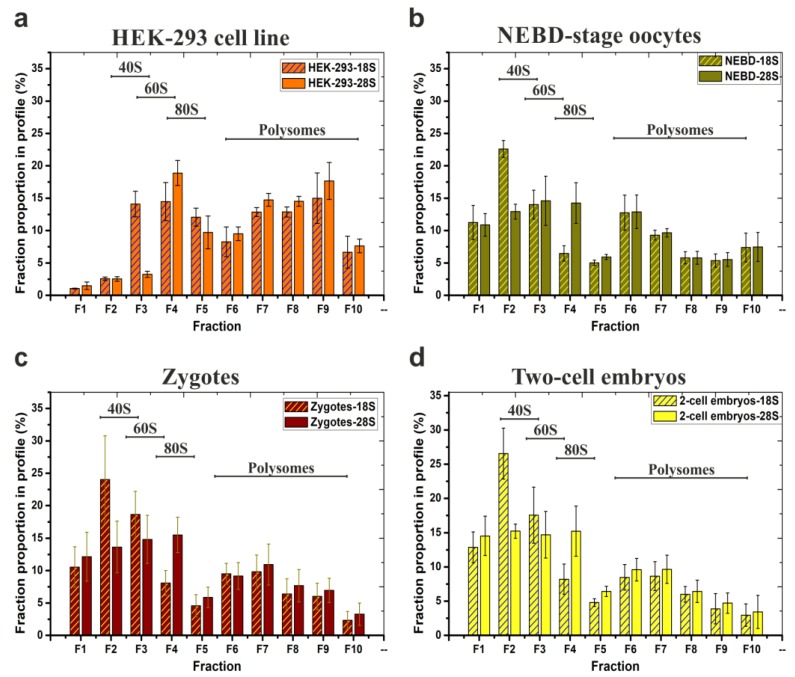
Typical distribution and abundance of ribosome- and polysome- ribonucleoprotein complexes in polysome profiles as determined by qRT-PCR analysis. Data are derived from four biological replicates of: (**a**) HEK-293 cell line (note, the same samples were analysed in Figure 2c); (**b**) 200 mouse NEBD-stage oocytes used for RNA-seq; (**c**) 200 zygotes and (**d**) 200 two-cell embryos. 18S and 28S contents in each fraction are displayed as percentages of the respective fraction compared to the whole polysome profile; error bars indicate ±SD of biological replicates. Note, qRT-PCR-based visualization of individual replicate/source polysome profiles are included in Appendix A.

**Figure 4 ijms-21-01254-f004:**
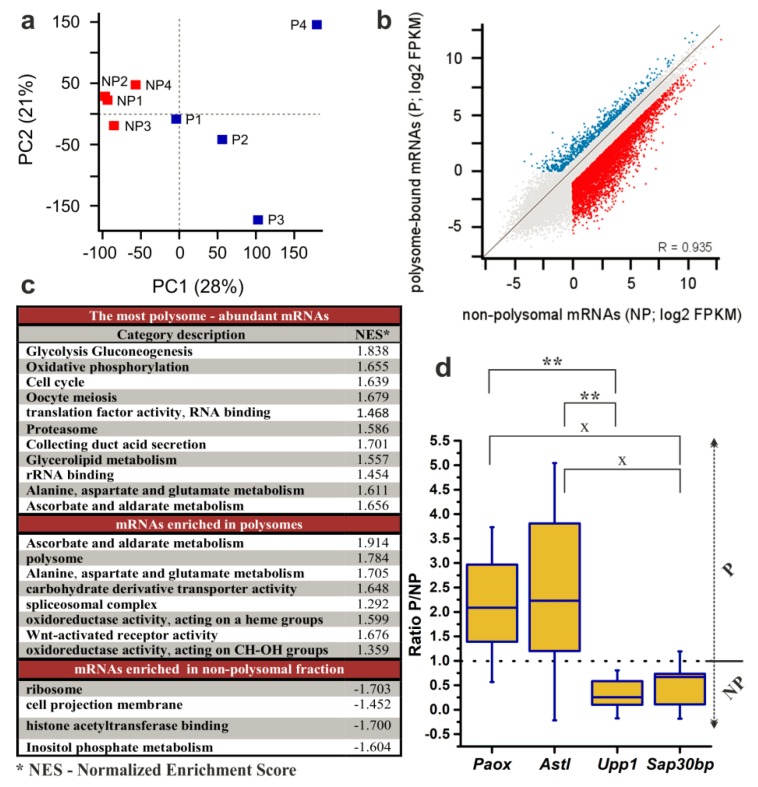
Analysis of RNA-seq results of 200 NEBD-stage oocytes. Non-polysomal (NP) and polysomal (P) RNAs were sequenced in four biological replicates. (**a**) Principal Component Analysis was applied to demonstrate variability among biological replicates. (**b**) Scatter plot highlighting genes with FPKM > 1 (fragments per kilobase per million reads mapped > 1) and > 2-fold enrichment in the NP (blue) and in the P (red). (**c**) Enriched gene categories according to GO (Gene Ontology) and KEGG (Kyoto Encyclopedia of Genes and Genomes) terms in dataset of all expressed genes with cut-off score of FPKM > 1. (**d**) qRT-PCR determination of the P/NP ratio for selected transcripts in three additional biological replicates. Box plot displays mean, 25th and 75th percentile and ±SD; dotted line indicates no fold change; ** *p* < 0.01; x signifies a lack of significance; Dunn´s multiple comparison is displayed only for pairs of transcripts that have been originally selected to be preferentially enriched in either the P or the NP dataset according to DeSeq2.

**Figure 5 ijms-21-01254-f005:**
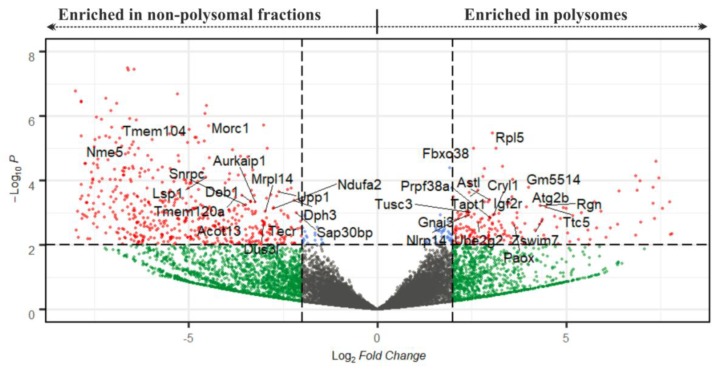
Volcano plot depicting transcripts according to their P/NP ratio after DeSeq2 analysis. The cut-off scores for the P/NP ratio (displayed as Log_2_ Fold Change) and the adjusted *p*-value (P*_adj_*, displayed as –Log_10_*P*) were set to 4 and 0.01, respectively. Red spots denote mRNAs that are significant in both parameters; cut-off scores are marked with dotted lines. Gene names highlight mRNAs with strong expression (with FPKM > 80 in at least one from P and NP datasets, Appendix A).

**Figure 6 ijms-21-01254-f006:**
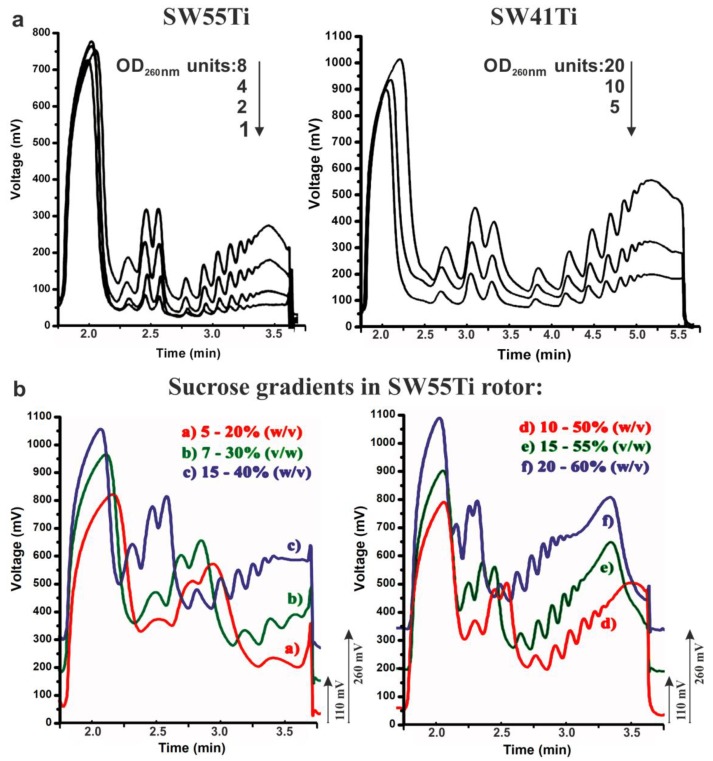
Sensitivity limit and versatility of polysome profiling in SW55Ti tubes. (**a**) 1–8 OD_260 nm_ of HEK-293 lysates were loaded onto 10%–50% sucrose gradients (in SW55Ti specific tubes) and centrifuged at 45,000 RPM for 65 min at 4 °C (left). Peak readability was compared to the one of conventional polysome profiles made in SW41Ti tubes, where 5–20 OD_260 nm_ were loaded onto 10%–50% gradients which were centrifuged at 35,000 RPM for 3 h at 4 °C (right). (**b**) Sucrose gradients of varying sucrose concentration ranges were prepared in SW55Ti tubes (a–f). Profiles in b + e and c + f were shifted up by 110 and 200 mV, respectively, to prevent overlapping of curve traces (to aid interpretation).

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
