# Peer review of "Identifying the Translatome of Mouse NEBD-Stage Oocytes via SSP-Profiling; A Novel Polysome Fractionation Method"

_ijms, 2020, doi:10.3390/ijms21041254_

Round 1

Reviewer 1 Report

This work by Masek and colleagues presents a new approach for polysome fractionation particularly useful for those samples for which a very limited amount of material is available; nonetheless, the approach can be also applied to more abundant samples.

In this specific manuscript, the authors apply the method to the characterization of actively translated transcripts in 200 mouse oocytes at the onset of nuclear envelope breakdown, confirming many previously reported findings and better characterizing the translatome of these specific cells.

In general, the work presented in this manuscript has the potential of being useful to those scientists studying translation and translation regulation, offering the advantages of allowing the analysis of scarce samples, saving time and materials.

However, there are some concerns that should be addressed:

Major comments

Comparing the rRNA copy number in cells of different types and species as a mean to assess the differences in loading is quite tricky. Indeed, these copy numbers can be heavily conditioned by the abundance of mature ribosomes themselves in the cytoplasmic lysate. To bypass this limitation, authors should include an experiment showing a polysomal profile and subsequent 18S/28S real time PCR analysis, also with a HEK 293 sample with an abundance comparable to that obtained by 200 oocytes. For the approach to be concretely useful also to other researchers, working with different samples than mouse oocytes, it is of crucial importance that the authors give information about the actual OD (or some other kind of quantification, like amount of proteins or RNA in the lysate) of the lysate of 200 oocytes. This information should be provided prior to fractionation. Similarly, authors could check the copy number of rRNAs also in lysate prior to fractionation, making sure that no rDNA contamination is present. Figures 2C and 3D do not match what reported in the text. More informative pictures should probably be representative of all the performed experiments (without a selection for Dunn’s comparison). Authors observe a decrease in polysomal fractions in zigotes and 2 cells embryos compared to NEBDs, and interpret it as a decrease in active translation. This conclusion cannot be drawn unless the authors test global protein synthesis, for example with nascent peptides labeling approach. At the beginning of paragraph 2.3 authors state that a comparison of polysome bound and non-polysome -bound mRNA pools produce almost identical results as comparison of polysome- bound mRNA against total RNA. This assumption has no fundament to the best of my knowledge, so this statement should be supported by a reference. Otherwise, the more appropriate approach for this kind of analysis is probably comparison of polysome-bound against total RNA, as generally done in this kind of studies. In line 283 authors state that Fraction 1 of polysome profile contains only tRNAs ad other small RNA particles. This statement is not supported by the pictures in the paper (fig. 3A, fig. 2B), where the presence of 18S and 28S is proven in fraction 1. In addition, transcript not recruited to polysomes likely co-sediment in this fraction. Therefore, the removal of this fraction from subsequent analyses is not correct, and analyses should be redone including it. In line 510 in the discussion section authors state that their method provides a unique snapshot of the currently synthesized proteins in NEBD-stage oocytes. This claim is not correct, since the abundance of polysomes-bound transcripts do not necessarily mirror the abundance of newly synthesized proteins (for instance, ribosomes could be stalling and no mature protein would be released). Therefore, this claim should be removed, unless supported by more definitive data.

Minor comments

Figure 1 upper panels: the arrows indicating h60S and y60S have been switched; Figure 1 lower panels: molecular weight markers should be added to the images; in addition, it is really weird that in the lanes corresponding to the first fractions of each profile the intensity of the bands does not reflect at all the height of the peaks. Maybe the material used to load these gels is not matched to the material used to show the profiles? Line 250: there’s some problem with this sentence, should be checked. Figure 5: this figure is not easily readable being so crowded. Maybe the authors could find a way to fix this? In addition, the legend does not match the figure (P/NP ratio between -2 and 2, but in the figure, hits are shown with values outside of this range) It is not clear why in figure 6 lysates with different RNA concentrations are used to load the gradients in SW55 and SW41? Can the authors explain? Also, could the authors explain why different lysate volumes are loaded on the different sucrose gradients in figure 6B? In figure S5, it would be extremely useful to detect ribosomal proteins (like one RPL and 1 RPS) in the same blots. There’s a number of typos that should be corrected throughout the text.

Reviewer 2 Report

In the study “Identifying the translatome of mouse NEBD-stage oocytes via SSP-profiling; a novel polysome fractionation method”, Masek et al demonstrated the usefulness of modified version of sucrose gradients for polysome profiling using various types of cells. The newly developed sucrose gradients method enables us to perform highly sensitive, reproducible, reliable polysome profiling, which is particularly valuable for experiments with scarce samples (unable to collect sufficient amounts for convenient sucrose gradients methods). They also provide usefulness of this method in combination with comprehensive methods such as RNA-seq for analyzing mRNA profiling in polysomal and non-polysomal fractions, that is translated and untranslated fractions. All experiments have been done with reproducible and reliable biological replicates and resulting data have been well analyzed and discussed. Since regulation of gene expression after transcription is extremely important for diverse biological processes including oocyte meiosis and development, this method is valuable for researches in many fields. I have only several comments for major and minor points as follows.

Major:

- Authors demonstrated the mRNA profiling using mouse oocytes at the stage after completion of NEBD, which seems to be prometaphase of meiosis I. However, reason why they chose this stage, not later stages of meiosis I or meiosis II, for mRNA profiling is unclear. Thereby, readers can not realize the importance of the data described by authors. Explanation for the importance to analyze mRNA profiling of this stage (not simply for the absence of the data in this stage) should be added.

Mainor:

- p.2, line 82: “fertilisable” would be “fertilizable”.

- p.3, line 108: The study of reference number 49 used zebrafish but not sea urchin. Add “zebrafish” in addition to sea urchin in this sentence.

- p.7, line 256: “NEDB” would be “NEBD”.

- p.16, line 596: The name of lysis buffer appeared here for the first time. The composition of this solution should be provided here, although it seems to be provided later (line 625).

Round 2

Reviewer 1 Report

Thank you very much for your revised version of the manuscript, and for the detailed rebuttal letter. 

The manuscript has improved and results are convincing. 

I appreciate your efforts very much.